# Syndromic Surveillance of Acute Gastroenteritis Using the French Health Insurance Database: Discriminatory Algorithm and Drug Prescription Practices Evaluations

**DOI:** 10.3390/ijerph17124301

**Published:** 2020-06-16

**Authors:** Frederic Bounoure, Damien Mouly, Pascal Beaudeau, Malek Bentayeb, Julie Chesneau, Gabrielle Jones, Mohamed Skiba, Malika Lahiani-Skiba, Catherine Galey

**Affiliations:** 1Laboratory of Pharmaceutical & Biopharmaceutical Technology, UFR of Health, Normandy University, Unirouen, 22 Bd Gambetta, 76183 Rouen Cedex, France; mohamed.skiba@univ-rouen.fr (M.S.); malika.skiba@univ-rouen.fr (M.L.-S.); 2DC2N, INSERM U1239, Unirouen, Normandy University, 76128 Mont Saint Aignan, France; 3Santé Publique France, French National Public Health Agency, F-94415 Saint-Maurice, France; Damien.MOULY@santepubliquefrance.fr (D.M.); pascal.beaudeau@santepubliquefrance.fr (P.B.); Malek.Bentayeb@kantarhealth.com (M.B.); julie.chesneau@santepubliquefrance.fr (J.C.); gabrielle.jones@santepubliquefrance.fr (G.J.); catherine.galey@santepubliquefrance.fr (C.G.)

**Keywords:** syndromic surveillance, acute gastroenteritis, waterborne outbreak, health insurance reimbursement database, drug prescription, France, pharmacy survey

## Abstract

The French national public health agency (Santé publique France) has used data from the national health insurance reimbursement system (SNDS) to identify medicalised acute gastroenteritis (mAGE) for more than 10 years. This paper presents the method developed to evaluate this system: performance and characteristics of the discriminatory algorithm, portability in mainland and overseas French departments, and verification of the mAGE database updating process. Pharmacy surveys with certified mAGE from 2012 to 2015 were used to characterise mAGE and to estimate the sensitivity and predictive positive value (PPV) of the algorithm. Prescription characteristics from these pharmacy surveys and from 2014 SNDS prescriptions in six mainland and overseas departments were compared. The sensitivity (0.90) and PPV (0.82) did not vary according to the age of the population or year. Prescription characteristics were similar within all studied departments. This confirms that the algorithm can be used in all French departments, for both paediatric and adult populations, with stability and durability over time. The algorithm can identify mAGE cases at a municipal level. The validated system has been implemented in a national waterborne disease outbreaks surveillance system since 2019 with the aim of improving the prevention of infectious disease risk attributable to localised tap water systems.

## 1. Introduction

Syndromic surveillance is defined by the Center for Disease Control (CDC) as an investigational approach where an existing automated data acquisition system is used for early outbreak detection or for the monitoring of disease indicators in real time or near real time [1]. Berger et al. described all the data sources that can be used for waterborne surveillance [2]. Drug sales have been used in many countries to detect acute gastroenteritis (AGE) outbreaks. Previous studies were either retrospective or predictive and analysed over-the-counter or prescription drug sales [3,4]. AGE is a very common syndrome typically defined as diarrhoea (three or more loose stools in 24 h) and/or vomiting [5]. AGE is used in epidemiology as a generic indicator for infections arising from faecal pathogens. It also provides good reactivity to environmental triggers (i.e., waterborne infectious disease), as AGE has a short incubation period. France has an extensive medical and administrative information system, which covers since 2016 the entire population living in France [6]. This drug reimbursement database, the “Système national des données de santé” (SNDS), collects information on prescriptions, patients, practitioners, and pharmacies from the National Health Insurance reimbursements database. Natural products or the over-the-counter drugs could be used for AGE treatment [7]; as they are not reimbursed, they are not taken into account in the SNDS database. In France, on average, 33% of AGE cases consult a general practitioner (GP) and almost all patients with medical consultation for AGE (medicalised AGE, mAGE) receive a treatment with reimbursed drugs (91%). Over-the-counter drugs are less used than reimbursed drugs with 8.9% and 30.9% of AGE cases, respectively [8].

Since 2007, the French national public health agency (Santé publique France) has been developing syndromic surveillance of mAGE based on the SNDS. A discriminating algorithm was developed to identify mAGE from the SNDS [9]. Data from the SNDS provide a daily exhaustive count of mAGE at the municipal level. The spatiotemporal characteristics of the mAGE indicator (day and municipality) are particularly suitable for the prevention of localised infectious disease risk attributable to tap water [10,11]. An automated detection system of waterborne disease outbreaks based on data from the SNDS and this algorithm has thus been deployed in France since 2019.

The aim of this paper is (i) to evaluate the mAGE discriminatory algorithm and its characteristics using independent data collected in pharmacy surveys; (ii) to compare the mAGE drug prescriptions obtained by using the algorithm on SNDS data from six French departments (mainland and overseas); and (iii) to describe how the updating process of the mAGE database is routinely checked at a national level for the most critical variables used in the algorithm.

## 2. Materials and Methods

### 2.1. Description of the mAGE Discriminatory Algorithm

The original algorithm was created based on a sample of GP prescriptions collected in pharmacies in 2000 and 2006, with associated medical diagnosis [9]. This sample was used to define relevant rules to differentiate mAGE prescriptions and non-mAGE prescriptions (intended for patients suffering from another disease). Characteristics included were the lag time between the prescription date and the day of delivery, the occurrence of non-mAGE specific drugs (non-mAGE drugs), mAGE specific drug combinations (mAGE drugs), and the duration of the treatment estimated from the amount of drugs prescribed (Figure 1). The algorithm criteria depend on the number of therapeutic classes for reimbursed mAGE drugs present on the prescription. These classes are intestinal absorbents, antipropulsives, antiemetics and antispasmodics, and rehydration salts. The algorithm is applied on a subset of prescriptions from the SNDS containing at least one of these mAGE drugs. Antispasmodics prescribed alone are excluded as they are widely prescribed. This reduces the volume of the subset of prescriptions without impairing the sensitivity for identifying mAGE. The list of non-mAGE drugs comprises specific drugs to treat other common digestive pathologies and drugs whose side effects include diarrhoea or vomiting [9]. Changes to the lists of mAGE and non-mAGE medications due to, for example, termination of reimbursement and removal from the market, could impact the performance of the algorithm. The impact is low if changes concern a medication inside a therapeutic class with drugs still reimbursed, as the reimbursed drugs will be prescribed instead. In the case of a new medication, the impact will be greater because these prescriptions would not be identified by the algorithm and extracted from SNDS. To address such risks, an update of the medications lists is completed annually, using the French medications database [12]. From 2011 to 2017, no major changes occurred in the mAGE therapeutic classes of drugs.

A mAGE case is defined as an individual identified through the SNDS as having a prescription containing at least one reimbursed mAGE drug recognised by the discriminatory algorithm (Figure 1 and Figure 2).

The algorithm also analyses the location of patient residence, GP, and dispensing pharmacy to distinguish resident from non-resident cases. A patient is considered a non-resident if they consulted a GP located more than 50 km from the place of residence. The municipality of the dispensing pharmacy is used when the information is not available for the GP.

The SNDS reimbursement database is updated monthly and is considered exhaustive at four months after prescription date, as 80% of mAGE prescriptions are identified at one month, 99% at two months, and 99.4% at three months [13]. Santé publique France currently updates its mAGE database every two months and has a 10-year history of mAGE cases.

### 2.2. Evaluation of the mAGE Discriminatory Algorithm: Pharmacy Surveys

GP prescription samples were collected annually, using the same method, from January to June, in two pharmacies in 2012 and in one pharmacy from 2013 to 2015 in the same geographical area (Seine Maritime department). Patients were enrolled if their prescription contained one of the defined mAGE drugs, except antispasmodics prescribed alone. All participants gave verbal consent. Participants were asked about their treatments, clinical symptoms, and medical diagnosis. Data collected from the prescription form included dates of medical examination and drug dispensation, drug names, and the number of boxes for each drug. Participants’ reported medical diagnoses were compared to diagnoses obtained from the algorithm applied to the prescription samples.

The performance of the algorithm was evaluated in terms of sensitivity and positive predictive value (PPV). Sensitivity evaluates the ability of the algorithm to correctly detect mAGE cases, and PPV is the probability that identified mAGE are true mAGE cases.

We evaluated the stability of the sensitivity and the PPV over time (2012–2015) and by age group (1–4 years old, 5–15 years old, and more than 15 years old), as well as the temporal stability of prescription characteristics (average number of therapeutic classes). The homogeneity of sensitivity and PPV across time and age classes was tested first by a global test for equality. When rejected (*p* < 0.05), two by two tests were performed using Holm correction for multi-testing. All analysis was performed on R version 3.4.3.

### 2.3. Comparison of mAGE Characteristics of Drug Prescription in Six French Departments

To evaluate the portability of the mAGE algorithm to different geographical areas, we reviewed the prescription characteristics in six (of 101) French departments: two mainland departments in the north and the south of France (Seine Maritime and Hérault) and four overseas departments (Martinique, Guadeloupe, Ile de la Réunion, and Guyane). The prescriptions extracted from the SNDS database from these departments were those dispensed in 2014 containing at least one mAGE drug (except antispasmodics). The mAGE algorithm was then applied and the mAGE and non-mAGE characteristics were studied for each department: mean number of drugs per prescription, lag time between the prescription date and the delivery day, percentage of mAGE cases, and the average number and frequency of the therapeutic classes on mAGE prescriptions.

### 2.4. Routine Check of mAGE Database Updating Process

Sante publique France updates the mAGE database every two months. Checks are performed on different variables to detect discrepancies or suspect changes in the rate of missing data, with a focus on location variables (the prescription is disregarded if the patient’s residence or the location of GP or dispensing pharmacy are not exploitable) and on the age of the patient (which if missing prevents the algorithm from processing the prescription). Patient, GP, and pharmacist locations are essential data for localised cluster detection as waterborne outbreak. Bias can occur in the use of mAGE for epidemiological purposes at a municipality geographical level if we do not distinguish non-resident from resident patients for mAGE. Data for routine checks from 2011 to 2017 at a national level are presented for these critical variables.

## 3. Results

### 3.1. Pharmacy Surveys

During the annual pharmacy surveys from 2012 to 2015, 1308 prescriptions were collected, including 728 mAGE (55%) and 580 non-mAGE (Table 1). These prescriptions were prescribed by 330 different GPs in the Seine Maritime department. Non-mAGE prescriptions covered a large set of situations: stockpile of medicines, prevention of drug side effects, gastroenteritic diseases including but not limited to abdominal pain, constipation, gastroesophageal reflux, or non gastroenteritic diseases such as flu, stress, and headache.

One-drug treatment was used in 8.1% of mAGE cases. Antipropulsives and antiemetics were the most used medications in case of a one-drug treatment. A treatment with two drugs was used in 45.6% of mAGE cases, including mostly antispasmodic, antiemetic, and antipropulsive drugs. A treatment with three drugs was used in 40.8% of mAGE cases, including mostly antiemetics, antispasmodics, and antipropulsives drugs. Intestinal absorbents and rehydration salts were not frequently used in the two-drug or three-drug treatments. A treatment with four drugs was rarely used (5.7%, data not shown). Almost 89% of mAGE cases purchased their drugs the day of consultation vs. 63% for other diagnoses.

A mAGE case prescription covered an average of 2.4 therapeutic classes of reimbursed drugs used to treat AGE (stable over the study period and for age groups) vs. 1.3 for non-mAGE.

The overall sensitivity and PPV of the mAGE algorithm were estimated to be 0.90 and 0.82, respectively. The annual sensitivity of the algorithm from 2012 to 2015 varied from 0.88 to 0.91, without significant differences. The annual PPV varied from 0.81 to 0.86, also without significant differences over the study period (Table 1). No difference was identified in the characteristics of prescription samples collected each year (number of prescribed drugs, frequency of class prescription, and lag time between prescription and dispensation (data not shown)).

Depending on the age group, the sensitivity varied from 0.87 to 0.96 and the PPV from 0.79 to 0.87. Global equality of the sensitivity according to age was rejected (*p* < 0.01). The sensitivity was significantly different (*p* < 0.05) between the 5–15-year-old and the >15-year-old age classes. No significant difference of PPV was identified between the three age groups.

### 3.2. Characteristics of mAGE and Non-mAGE Case Prescriptions from the SNDS across Six French Departments

More than 1 million prescriptions were extracted from the SNDS in 2014 for the six departments included in the study (Table 2). Around 45% were for mAGE, with an average of 2.2 prescribed drugs. The delay between prescription date and dispensation day was comparable for the two mainland departments, with approximately 92% of mAGE cases filling the prescription on the same day versus 68% (average delay of 3.6 days) for non-mAGE cases. The delays were similar in overseas departments with more than 93% of prescription and delivery on the same day for mAGE versus 75% (average of 2.7 days) for non-mAGE. The average number of therapeutic classes was similar across all departments for mAGE (2.1 to 2.3) and non-mAGE (1.2 to 1.3). These values were close to values obtained from the pharmacy surveys, which identified 55.6% of prescriptions for mAGE with an average of 2.4 prescribed drugs, and 89% of mAGE cases’ prescriptions dispensed on the same day as GP visit (Table 1).

The most prescribed therapeutic classes were also similar between the SNDS analysis in six departments and the pharmacy surveys with a prescription frequency depending on department. The drug classes most commonly used were the same: antiemetics, with an average prescription frequency of 65.5% and antipropulsives with 59.8%, vs. 73.3% and 77.2%, respectively, in pharmacy surveys. Antispasmodics were found on 55.1% of prescriptions and adsorbents on 30.3% of the prescriptions vs. 54.3% and 25.6%, respectively, in pharmacy surveys. The prescription of rehydration salts varied to a greater degree by department, with the highest use in Guyane (17.6% vs. 10% for others). These similar results across departments seem to confirm the relevance of the algorithm for use in overseas departments.

### 3.3. Routine Check of mAGE Database Updating Process

Between 2011 and 2017, 13,300,000 to 17,100,000 prescriptions containing at least one drug prescribed for AGE were extracted yearly from the SNDS for all 101 French departments (Table 3). The algorithm was run on almost all the extracted prescriptions, as the prescription and dispensation dates were always completed in the SNDS, and the proportion of missing data for patient age decreased from 0.2% to 0 over the study period. The proportion of prescriptions for mAGE ranged from 36% to 45%, depending on the year.

Municipality codes are essential for the geolocation of the mAGE case and their categorisation as a local resident or non-resident (defined as patients who consulted the doctor farther than 50 km from home). “Fully located case” means that both practitioner’s and patient’s residence location are available. This condition makes it possible to distinguish “resident” patients, with the hypothesis that they are exposed in their area of residence, from “non-resident” patient, for whose exposure likely occurred away from home. The pharmacy location was used when the GP’s municipality was missing. A sharp decrease in missing geolocation data was observed from 2011 to 2017, as a result of a decrease of the missing data for the patient. Since 2012, 96% of mAGE cases can be classified as a resident or as a non-resident, and in 2016 this proportion increased to 98%.

## 4. Discussion

### 4.1. Main Results

The evaluation presented in this study based on data collected in pharmacy surveys demonstrated that our discriminatory algorithm was able to effectively distinguish mAGE from prescriptions for other pathologies. The sensitivity and PPV of the algorithm are high, around 0.90 and 0.82, respectively. This is a robust indicator for surveillance of AGE, as it remains stable over both time and in different geographical areas. The evaluation using data from the SNDS, demonstrated that most of the parameters remained stable across French departments: the percentage of mAGE cases with prescriptions containing at least one drug to treat AGE (between 40% and 50%), mean number of therapeutic classes for mAGE cases (2.2), percentage of mAGE prescriptions prescribed and dispensed within 24 h (over 90%). Only therapeutic class proportion on mAGE prescriptions seems department dependent. As the mAGE discriminating algorithm is based on syndromic treatment, it is not possible to estimate whether differences are due to specific pathogen distribution and/or GP prescription practices between departments.

### 4.2. Surveillance System Representativeness

Regarding the use of data from the SNDS, only mAGE cases that have consulted and filled prescriptions for AGE reimbursed drugs are detected. Van Cauteren et al. [8] showed, using a population-based retrospective cross-sectional telephone survey, that 33% of AGE cases consulted a GP, and 31% bought the prescribed drugs. However, GP consultation is dependent on age [14], the nature of the pathogen and the duration and severity of the symptoms [15], and access to health services [16]. Cohort investigations of waterborne outbreaks have shown that the proportion of patients consulting for AGE varies greatly: from 80% in an area where drinking water was highly contaminated by *Cryptosporidium* oocysts [17], to 10% in a campground serviced by drinking water with probable viral contamination [18].

The SNDS covers 99% of the population living in France [7]. However, caution is required when studying certain subpopulations, such as the elderly or students. Elderly persons living in retirement homes may not appear in the SNDS if drugs are provided by a hospital pharmacy. Their proportion increases with age, especially after 85 years old. Students are mobile and often prefer to report to administrations at their parent’s address instead of their own. They may therefore be classified as non-resident when they in fact consult in their current (but unregistered) municipality of residence.

### 4.3. Limitations

There are several limitations to this work. First, possible representativeness bias could be introduced in pharmacy surveys, as data were all collected from January to June. However, this bias is probably limited, as this period covers both the winter AGE epidemic periods and non-epidemic periods. Second, regarding the algorithm, certain populations are not fully covered. For example, infants under one year of age are not taken into account. The prescriptions for mAGE in infants frequently contain rehydration salts. This medication can also be systematically prescribed upon discharge from the maternity ward to promote its future use to cope with the risk of mAGE-associated dehydration. The sensitivity of the algorithm for the elderly is lower, as their prescriptions often contain several drugs for various chronic diseases, and thus, may be wrongly rejected by the algorithm.

### 4.4. External Comparison

Comparisons between the results of this work and previous studies related to AGE demonstrate similar findings. A population-based survey estimated an annual incidence rate of 0.33 AGE cases/person-year in the French mainland population between May 2009 and April 2010, with 31% of AGE cases filling a prescription from a GP [8]. The 2010 average incidence of mAGE (0.10 cases/person-year) drawn from SNDS data is consistent with the survey results (0.33 × 0.31 = 0.10). An algorithm derived from our algorithm was also tested on a commercial SNDS-like database, known as the Longitudinal Treatment Dynamics™ (LTD) database, directly fed by 30% of French pharmacists. For the latter study, drugs for peptic ulcer and gastro-oesophageal reflux disease were excluded as they were too frequently prescribed. This difference reduces the specificity (PPV) in comparison of our algorithm. The authors have found a strong agreement in the dynamic of mAGE activity in winter, between the estimate derived from the LTD data and the reporting of a primary care surveillance system (GP Sentinelles network) with a correlation coefficient between 0.84 and 0.94 on the mAGE weekly rate [19]. The work of Vilcu et al. covers an estimation of regional or national mAGE rate during winter seasons with an exclusion of overseas departments and is not suitable for a local mAGE surveillance.

### 4.5. Perspectives for Waterborne Infectious Disease Surveillance

mAGE syndromic surveillance based on SNDS data is particularly relevant for studying and preventing waterborne infectious risk of faecal origin [11]. Data from previously investigated AGE outbreaks provided the opportunity to retrospectively test the SNDS database for outbreak descriptions [14,20,21]. Mouly et al. compared data from the SNDS and data from cohort studies obtained during two waterborne infections and demonstrated that the temporal distribution of cases, the day of the peak, and the duration of the epidemic were similar [14].

The SNDS approach has also been used to evaluate outcomes of water safety plans (WSPs) implemented in two large drinking water systems in France [22], and to evaluate the endemic risk of AGE based on drinking water conditions in French urban areas in time series studies [23].

In 2019, a waterborne outbreak surveillance system based on the presented algorithm was implemented in France. The objectives of this system are to facilitate the identification and management of drinking water systems that need to be secured to protect consumers’ health and to improve the prevention of waterborne disease outbreak. This surveillance system is the result of evaluation of different methods of detection of mAGE clusters with a possible waterborne origin [10,24,25], followed by a pilot study in seven French departments to test the feasibility of such a surveillance system before any national implementation [26]. It should increase 10 to 100-fold the number of waterborne outbreaks reported to health authorities [24,26]. The current surveillance system uses a web-application also developed by Santé publique France (EpiGEH) (article in process) for data presentation and is based on the localisation and the characterisation of clusters of mAGE cases sharing the same water distribution system [10]. The system includes local environmental investigations to identify water distribution system contaminations, failures, or vulnerability and aims to aid in prevention strategies and WSPs for water distribution systems.

## 5. Conclusions

Santé publique France has developed a novel syndromic surveillance system of mAGE based on a discriminatory algorithm that exploits data from the SNDS medication database. SNDS data are routinely and automatically available and cover the whole French population. The algorithm used to identify prescriptions for mAGE has been evaluated and validated on data from different mainland and overseas French departments and for both paediatric and adult populations. The implementation of the algorithm to develop a nationwide surveillance system for retrospective waterborne outbreak surveillance highlights the utility/benefit of using the SNDS database for syndromic surveillance of mAGE.

## Figures and Tables

**Figure 1 ijerph-17-04301-f001:**
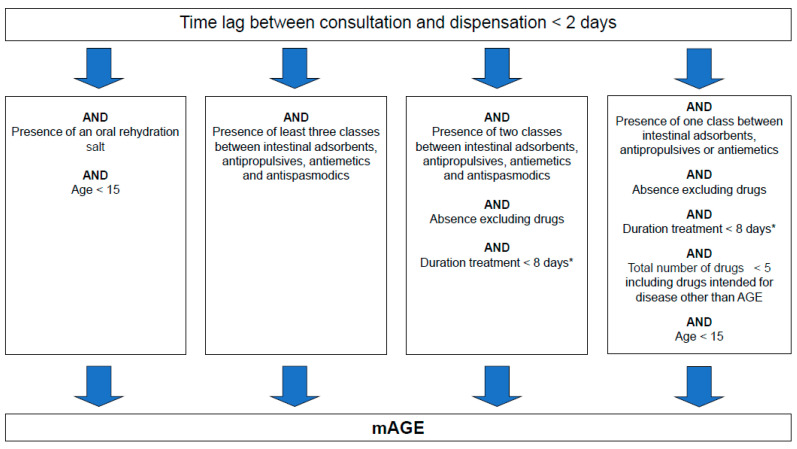
Algorithm used for medicalised acute gastroenteritis (mAGE) case discrimination based on drug reimbursement data. * Estimated from both the content and the number of the boxes dispensed of any mAGE drugs.

**Figure 2 ijerph-17-04301-f002:**
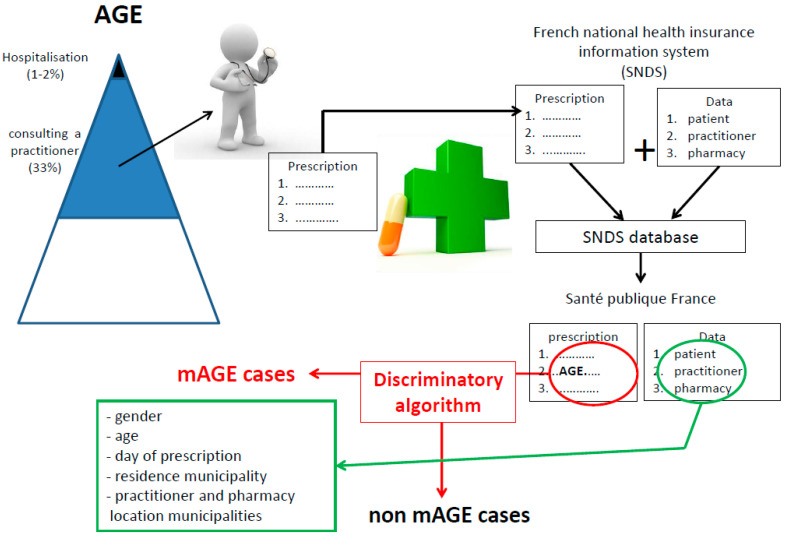
Process and information collected for mAGE from the SNDS database (French national health insurance reimbursement system).

**Table 1 ijerph-17-04301-t001:** Characteristics of Pharmacy survey prescriptions (*n* = 1308) and sensitivity, positive predictive value of the mAGE algorithm evaluated for stability over time and on age (*n* = 1178, no data for 2015).

	Number of Prescriptions						Average Number of Therapeutic Classes to Treat AGE **	Algorithm Performance	
	All	%	mAGE	%	Non-mAGE	%		Sensitivity	PPV
Year									
2012	700	53.5%	413	57.1%	287	42.9%	2.5	0.91	0.81
2013	169	12.9%	129	76.3%	40	23.7%	2.4	0.88	0.86
2014	309	23.6%	120	38.8%	189	61.2%	2.4	0.87	0.83
2015	130	10%	66	50.7%	64	49.3%	2.4	0.91	0.82
All	1308	100%	728	55.6%	580	44.4%	2.4	0.90	0.82
Age group									
1–4 years old	178	15.1%	125	70.2%	53	29.8%	2.3	0.89	0.79
5–15 years old	184	15.6%	153	83.1%	33	26.9%	2.4	0.96 *	0.87
>15 years old	826	70.1%	384	46.5%	430	53.5%	2.5	0.87	0.81
All	1178	100%	662	56.2%	516	43.8%	2.4	0.90	0.82
Characteristics of prescriptions									
Therapeutic classes on prescriptions	**mAGE**		**%**		**Non-mAGE**	**%**			
Intestinal antispasmodics	395		54.3%		137	23.6%			
Antiemetics	535		73.3%		269	46.4%			
Intestinal antipropulsive	562		77.2%		253	43.7%			
Intestinal absorbents	186		25.6%		111	19.1%			
Oral rehydration salts	56		7.6%		8	1.4%			
AGE drug combinations									
One-drug treatment	58		8%		385	66.3%			
Two-drugs treatment	332		45.6%		158	27.2%			
Three-drugs treatment	297		40.8%		34	6%			
Four-drugs treatment	41		5.6%		3	0.5%			
Prescription and dispensation on the same day	646		88.7%		367	63.3%			

** Only for mAGE prescriptions. * Significant difference (*p* < 0.05). PPV: positive predictive value.

**Table 2 ijerph-17-04301-t002:** Characteristics of medicalised acute gastroenteritis (mAGE) and non-medicalised AGE (non-mAGE) prescriptions in 2014 (1,058,323 prescriptions from the SNDS).

Department	Hérault	Seine Maritime	Guadeloupe	Martinique	Guyane	La Réunion
Number of prescriptions	271,346	311,822	57,883	57,776	38,681	320,815
% mAGE	41.0%	43.5%	45.4%	43.0%	45.4%	50.1%
Mean number of therapeutic classes						
mAGE	2.27	2.27	2.10	2.18	2.18	2.31
Non-mAGE	1.27	1.24	1.20	1.20	1.24	1.21
Prescription and dispensation on the same day (%)						
mAGE	91.2%	91.3%	93.1%	93.1%	93.2%	96.9%
Non-mAGE	67.5%	68.5%	75.3%	78%	80.8%	86%
Therapeutic classes on mAGE prescriptions						
Intestinal antispasmodics	57.9%	57.4%	53.5%	59.0%	46.3%	55.6%
Antiemetics	65.6%	68.8%	63.0%	66.4%	67.3%	61.6%
Intestinal antipropulsive	62.2%	64.3%	55.7%	49.6%	65.6%	61.5%
Intestinal absorbents	31.6%	27.8%	25.8%	33.4%	21.5%	41.5%
Oral rehydration salts	10.0%	8.8%	12.2%	9.6%	17.6%	10.5%

**Table 3 ijerph-17-04301-t003:** Number of prescriptions extracted from the reimbursement database (SNDS) and classified as related to medicalised acute gastroenteritis (mAGE) cases. 2011–2017, all French departments.

	2011	2012	2013	2014	2015	2016	2017
Prescription extracted from SNDS							
Total extracted (number)	15,953,058	17,141,884	14,880,301	15,855,962	15,864,282	15,921,223	13,365,121
Missing age data (number)	24,179	24,009	18,048	15,082	8908	0	0
Missing age data (%)	0.2%	0.1%	0.1%	0.1%	0.1%	0.0%	0.0%
mAGE							
Total number of mAGE	6,970,559	7,020,943	6,624,724	6,041,682	5,697,805	6,078,354	5,008,460
% of extracted prescriptions	44%	41%	45%	39%	36%	38%	37%
Missing geolocation							
Patient’s residence	4.4%	3.2%	3.0%	3.0%	2.1%	1.4%	1.7%
Practitioner’s location	6.1%	6.3%	6.7%	7.1%	7.6%	7.7%	9.0%
Practitioner’s and pharmacist’s locations	0.2%	0.2%	0.2%	0.2%	0.1%	0.1%	0.1%
Case location classification impossible	4.6%	3.4%	3.2%	3.1%	2.2%	1.4%	1.8%

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
