# Peer review of "Syndromic Surveillance of Acute Gastroenteritis Using the French Health Insurance Database: Discriminatory Algorithm and Drug Prescription Practices Evaluations"

_ijerph, 2020, doi:10.3390/ijerph17124301_

Round 1

Reviewer 1 Report

Revisions are required.

  1. A brief introduction to SNDS, such as its function and operation, may be given, so that the readers can better understand the study background.
  2. Natural products, or herbal medicines, are commonly used for treatment of acute gastroenteritis, or symptoms of acute gastroenteritis such as diarrhea. The authors may read and cite some references in this field, for example, PMID: 30555015 and PMID: 28844213. The reviewer suggests that it should be analyzed whether prescriptions of natural products or herbal medicines can be identified by the algorithm and extracted from SNDS. And if not, this point should be discussed as a study limitation.
  3. It is required to provide a table to compare the authors’ results with others, may be not in the French. The data from PMID: 27105415 should be included.

Author Response

We thank you for your useful comments to improve the article.

1. We have added more details on the SNDS database to explain that SNDS includes only reimbursed drug prescribed by a general practioner and not over the counter drugs or natural products. These SNDS data are provided by National Health Insurance NHI. (line 44 to 49)

2. We have mentioned that the natural products could be used for the AGE treatment and added a suggested publication. In France, the natural products are not included in the SNDS database because they are not reimbursed by NHI. We have added some data on the use of the over-the-counter drugs (line 44 to 49)

3. We have detailed a little more studies performed by SpF on the waterborne disease outbreak surveillance system that is implemented since 2019 in whole France, using NHI data. Such a system is going to improve the waterborne outbreak detection associated with tap water consumption in France. We have added the ref PMID: 2710541 as suggested and two others. More results concerning the waterborne detection with EpiGEH is the aim of another paper under process. (line 286 to 295)

Reviewer 2 Report

Editors and Authors

Thank you for the opportunity to review this manuscript. 

Overall I found this article interesting pertaining to tracking waterborne outbreaks. The algorithm use was supported by the evaluation conducted in this study. 

The content is relevant and the information accurate as far as I can tell. The references support the study. 

This study is logically presented. 

I found this algorithm informative. The tables were presented with the information needed; as well the information in the tables supported the discussion and conclusion. 

Please include a specific paragraph addressing the implications. While it is briefly described in the abstract, it could be presented more clearly in the conclusion or have its own paragraph prior to the conclusion. 

Author Response

We thank you for your useful comments to improve the article.

Implications of the use of medicalised acute gastroenteritis (mAGE) from the national health insurance reimbursement system (SNDS) are now more detailed in paragraphe 4.5 (line 286 to 295). We have mentioned more studies performed by SpF on the waterborne disease outbreak surveillance system that is implemented since 2019 in whole France, using SNDS data. We have added 3 new references. Such a system is going to improve the waterborne outbreak detection associated with tap water consumption in France. The major implication (EpiGEH) is the aim of a specific publication under process.